# Antimycotic effect of 3-phenyllactic acid produced by probiotic bacterial isolates against Covid-19 associated mucormycosis causing fungi

**Ravikumar Lunavath**[1], **Saddam Hussain Mohammad**[1], **Kiran Kumar Bhukya**[1], **Anuradha Barigela**[1], **Chandrasekhar Banoth**[1], **Anil Kumar Banothu**[2], **Bhima Bhukya**[1]*

**1** Centre for Microbial and Fermentation Technology, Department of Microbiology, University College of Science, Osmania University, Hyderabad, India, **2** Department of Veterinary Pharmacology and Toxicology, College of Veterinary Science, P. V. Narsimha Rao Telangana Veterinary University, Rajendranagar, Hyderabad, India

* bhima.ou@osmania.ac.in

**Data Availability Statement:** All relevant data are within the manuscript.

## Abstract

The Covid-19 associated mucormycosis (CAM) is an emerging disease affecting immuno-compromised patients. Prevention of such infections using probiotics and their metabolites persist as effective therapeutic agents. Therefore, the present study emphasizes on assessment of their efficacy and safety. Samples from different sources like human milk, honey bee intestine, toddy, and dairy milk were collected, screened and characterized for potential probiotic lactic acid bacteria (LAB) and their metabolites to be used as effective antimicrobial agents to curtail CAM. Three isolates were selected based on probiotic properties and characterized as *Lactobacillus pentosus* BMOBR013, *Lactobacillus pentosus* BMOBR061 and *Pediococcus acidilactici* BMOBR041 by 16S rRNA sequencing and MALDI TOF-MS. The antimicrobial activity against standard bacterial pathogens showed >9 mm zone of inhibition. Furthermore, the antifungal activity of three isolates was tested against *Aspergillus flavus* MTCC 2788, *Fusarium oxysporum*, *Candida albicans* and *Candida tropicalis* where the results showed significant inhibition of each fungal strain. Further studies were carried out on lethal fungal pathogens like *Rhizopus* sp. and two *Mucor* sp. which are associated with post Covid-19 infection in immunosuppressed diabetic patients. Our studies on CAM inhibitory effect of LAB revealed the efficient inhibition against *Rhizopus* sp. and two *Mucor* sp. The cell free supernatants of three LAB showed varied inhibitory activity against these fungi. Following the antimicrobial activity, the antagonistic metabolite 3-Phenyllactic acid (PLA) in culture supernatant was quantified and characterized by HPLC and LC-MS using standard PLA (Sigma Aldrich). The isolate *L. pentosus* BMOBR013 produced highest PLA (0.441 g/L), followed by *P. acidilactici* BMOBR041 (0.294 g/L) and *L. pentosus* BMOBR061 (0.165 g/L). The minimum inhibitory concentration of HPLC eluted PLA on the *Rhizopus* sp. and two *Mucor* sp. was found to be 180 mg/ml which was further confirmed by inhibition of total mycelia under live cell imaging microscope.

**Funding:** This work was financially supported by the University Grants Commission (UGC), Govt. of India through CSIR-UGC fellowship (Ref. No: 21/06/2015(I)EU-V), the Ministry of Human Resource Development, Govt. of India through RUSA 2.0 program and Department of Science and Technology, Govt. of India through DST-PURSE program. The funders had no role in study design, data collection and analysis, decision to publish, or preparation of the manuscript.

**Competing interests:** The authors have declared that no competing interests exist.

## Introduction

Many broad range opportunistic mycotic infections associated with Coronavirus disease 2019 (Covid-19) are caused by mainly fungal pathogens as co-infections in Covid-19 patients. In recent years, there has been a large number of cases reported worldwide particularly in India, a life-threatening fungal infection in Covid-19 patients admitted to hospitals known as 'Mucormycosis' [1]. It is commonly referred to as 'Black fungus' and its occurrence has increased rapidly in the second wave when compared to the first wave of Covid-19 pandemic in India [2]. Mucormycosis is an angioinvasive infection caused by saprophytic fungi belonging to the genera *Rhizopus*, *Mucor*, *Rhizomucor*, *Absidia*, *Cunninghamella*, *Apophysomyces*, and *Saksenaea* of Order Mucorales [3]. Globally, *Rhizopus arrhizus* is the most commonly isolated organism from patients in all clinical forms of mucormycosis. It is responsible for approximately 60–70% of mucormycosis cases in humans and also accounts for 90% of the rhino-cerebral form of infection [1, 4]. The mucormycosis is a rare but lethal fungal disease, more aggressive, and commonly infect immunosuppressed patients through inhalation of fungal spores [1, 5]. The fungal spores can infiltrate blood vessels and spread to brain and other vital organs via bloodstream, result in causing infection [3]. The occurrence is more common in patients who are immunosuppressed due to diabetes mellitus (diabetic ketoacidosis form), neutropenia, thrombocytopenia, cancer, higher levels of serum iron, and patients with organ transplantation. As a result, the number of cases is increased with higher susceptibility [4]. Globally, Covid-19 with diabetes mellitus is a major risk factor for developing mucormycosis, with an overall mortality rate of 46% [1]. The hallmark of mucormycosis infection is necrosis of host tissue because of extensive angioinvasion, thrombosis, obstruction of blood vessels, and also, the absence of necrotic eschar which does not preclude the diagnosis of infection [6, 7]. In India, uncontrolled diabetes is the most common underlying disease linked with mucormycosis and the prevalence of this secondary infection is approximately 70 times higher compared with global data, with a mortality rate ranging between 28–52% [1, 8].

Gastrointestinal mucormycosis can be acquired through ingestion of contaminated foods or food products like fermented milk and bakery products introduced to the gastrointestinal tract (GIT) and the infection is characterized by fungal invasion into gut mucosa, submucosa, and blood vessels. It accounts for <10% of total cases of mucormycosis, one-third of the cases occur in infants and children which show a higher mortality rate (85%) due to the high occurrence of bowel perforation and therefore establishment of diagnosis is difficult [5].

Probiotics are nonpathogenic, live microorganisms (yeast or bacteria) that when ingested in sufficient quantity confer health benefits to the host [9]. Most probiotics belong to the lactic acid bacteria (LAB) including various species of *Lactobacillus*, *Pediococcus*, *Leuconostoc*, *Lactococcus*, and *Streptococcus* has acknowledged the Qualified Presumption of Safety (QPS) consideration by the European Food Safety Authority (EFSA) [10, 11]. The lactic acid bacteria have generally regarded as safe (GRAS) status approved by the Food and Agricultural Organization of the United States (FAO) [10]. Several health benefits associated with human health is conferred by probiotic bacteria, which include reducing the bowel disease, improvement of the normal microbiota and lactose utilization, prevention of antibiotic-associated diarrhea, anti-carcinogenic, anti-cholesteric effect, and increase immune response [12, 13]. Probiotics are known to diminish the severity of GIT and the upper respiratory tract infections by enhancing the innate and adaptive immune systems [14]. Probiotics have been found to employ health benefits via numerous potential mechanisms including clamping down the pathogens, mmunomodulation, maintenance of intestinal barrier function, and modulation of epithelial cell signal transduction [15]. Strikingly, probiotics have been shown to exert strong antimicrobial effects via neutrophils, alveolar macrophage, natural killer cells and elevated levels of pro-

inflammatory cytokines like TNF and IL-6 in the lung [14]. Probiotic supplements could be crucial for maintaining an optimal immune system in Covid-19 and its associated opportunistic secondary infections [16]. Additionally, probiotic supplementation could be helpful for tackling CAM/mucormycosis and restoring immunological homeostasis [17]. A wide variety of antimicrobial metabolites are produced by LAB including lactic acid, acetic acid, phenyllactic acid (PLA), hydrogen peroxide, short-chain fatty acids, 4-hydroxy phenyllactic acid, reuterin, cyclic dipeptides, and bacteriocin [18–20].

The 3-Phenyllactic acid (2-hydroxy-3-phenylpropanoic acid, PLA) and its derivative hydroxyphenyllactic acid (HPLA) are organic acids produced by various LAB species, mainly responsible for antifungal activity but their production is strain-dependent [21]. The PLA is an antagonistic compound with a broad-spectrum inhibitory activity against a range of Gram-positive and Gram-negative bacteria. In addition, it inhibits fungi such as yeasts (*Candida* and *Rhodotorula* spp.), a wide range of molds (*Aspergillus* and *Penicillium* spp.) [19, 22]. PLA is responsible for antifungal activity which helps in prolonging the shelf life of foodstuffs, it can diffuse easily into the food and feed due to hydrophilic nature. Also, PLA is used as skin protecting agent to reduce the skin wrinkles, and improve the meat quality when used as feed additive [23]. The PLA is used as a pharmaceutical agent with a great potential for the treatment of acute myocardial infarction and inhibition of platelet aggregation since its structural analogue 'Danshensu' (Salvia miltiorrhiza) was in practice from Chinese medicinal herb [24].

The current study focuses on the evaluation of probiotic and antimycotic efficiency of *Lactobacillus pentosus* BMOBR013, isolated from human milk, *Lactobacillus pentosus* BMOBR061, and *Pediococcus acidilactici* BMOBR041 isolated from buffalo milk.

## Materials and methods

### Sample processing and isolation of lactic acid bacteria

A total of 33 samples were collected from different sources like honey bee intestine, soil, toddy, milk from volunteer lactating mothers (Institutional Ethics Committee, Institute of Genetics and Hospital for Genetic Diseases, Osmania University. No. 327/IOG/IECBR/2022; Participants were informed about the study and taken written consent), raw milk from cow, goat and buffalos (Institutional Ethics Committee No. 40/25/CVSc, Hyd. IAEC, P.V. Narsimha Rao Telangana Veterinary University, Hyderabad), and transported to the laboratory under sterile conditions at 4˚C. Aliquots of 0.1mL 10-fold diluted samples were inoculated on de Man, Rogosa, and Sharpe (MRS-GM369, HiMedia, India) agar containing 0.5% (w/v) calcium carbonate (CaCO3-RM1044, HiMedia, India) and plates were incubated under anaerobic conditions at 37˚C for 24–48 h. Morphological distinct colonies were selected randomly from MRS agar plates andexamined for morphology, gram staining, and catalase production. Finally, gram positive and catalase negative pure cultures were cultured in MRS agar slants and stored at 4˚C for further use. Glycerol stock (20% glycerol) of pure cultures was maintained at -80˚C for further assays.

### Microorganism and media

Bacterial strains *Pseudomonas aeruginosa* (ATCC 27853), *Staphylococcus aureus* (ATCC 49134), *Escherichia coli* (ATCC 8739), *Bacillus subtilis* (ATCC 23857), *Salmonella typhi* (ATCC CVD 909 (202117)), and *Klebsiella pneumonia* (ATCC 33499) were procured from the American type culture collection (ATCC). *Aspergillus flavus* MTCC 2788, *R. arrhizus* MTCC 24794 were procured from the Microbial type culture collection (MTCC), Pune, India. *Rhizopus* sp. and two *Mucor* sp. isolates of nasal sinuses of post Covid-19 mucormycosis infected

patient samples procured from NRI Medical College and Hospital, AP). The other fungal cultures like *Candida albicans* (from patient sample), *Candida tropicalis* MTCC 184, and *Fusarium oxysporum* KACC42109 were used in this study. Yeast strains were cultured in yeast extract peptone dextrose agar (YPD-G038, HiMedia, India) and molds in potato dextrose agar (PDA-M096, HiMedia, India) at 28°C for 3–5 days and stored at 4°C. Spore inoculum was prepared by culturing the molds on PDA plates until sporulation occurs, and the spores were collected by vortexing with sterile distilled water containing tween 80 (0.05%) (LQ520X, HiMedia, India). The spore concentration was determined using a haemocytometer and adjusted to $10^5$ spores/ml.

## *In vitro* probiotic characteristics of LAB isolates

**Tolerance to virtual gastric and bile juice.** Tolerance to simulated gastric juice and bile was evaluated with slight modifications in plate count method [25]. The simulated gastric juice was prepared by dissolving 1% pepsin (w/v) (RM084, HiMedia, India) in MRS broth and pH was adjusted to 2.5 with 5.0 mol/l hydrochloric acid. Ten mL each of overnight grown bacterial cultures was harvested by centrifugation at 10,000 rpm, for 10 min and pellets were washed twice with PBS buffer. Two mL of simulated gastric juice was added to each pellet and 100 μl of suspension was withdrawn at 0, 1, and 2 h of incubation, diluted, inoculated on MRS agar plates and incubated at 37°C for 48 h. Similarly, for bile tolerance, 2 mL each of bile solution consisting 1.28 g/l NaCl (MB023, HiMedia, India), 0.239 g/l KCl (MB043, HiMedia, India), 6.4 g/l NaHCO₃ (MB045, HiMedia, India) and 0.3% (w/v) ox bile (RM621, HiMedia, India) was added to bacterial cell pellets. Bile solution suspension was withdrawn at 0, 1, and 2 h of incubation, diluted and 100 μl of a suitable dilution was inoculated on MRS agar plates and incubated at 37°C for 48 h. After incubation, the colonies on the plates were counted. Simulated gastric juice and bile tolerance was evaluated by counting the number of viable colonies, and each strain was carried out in triplicate. The probiotic strain *L. acidophilus* NCIM 2285 was used as a referral strain.

**Tolerance to temperature and sodium chloride.** The ability of LAB isolates to grow at different temperatures and in different NaCl concentrations was assessed as previously described [26]. In brief, 1% (v/v) overnight grown LAB cultures were inoculated in 10 mL each of MRS broth tubes containing bromocresol purple indicator with different concentrations of NaCl (w/v) (4, 6.5, and 9.5%) and incubated at 37°C for 24 h. Similarly, MRS broth tubes containing bromocresol purple indicator were inoculated with 1% LAB cultures and incubated for 24 h at different temperatures (20, 30, 37, 45, 50, 55 and 60°C). After incubation, MRS broth tubes were observed for color change from purple to yellow. The strain *L. acidophilus* NCIM 2285 was used as referral probiotic strain.

**Phenol tolerance.** The ability of LAB isolates for tolerance to phenol was assessed according to previously described method [26]. In brief, 1% (v/v) overnight grown probiotic culture inoculum was inoculated in 10 mL sterile MRS broth tubes with specific concentrations of phenol (0.1–0.4%) (P1037, Sigma-Aldrich) and incubated at 37°C for 24 h. After incubation, growth was observed by measuring OD$_{620}$. The probiotic strain *L. acidophilus* was used as referral strain.

**Bacterial cell surface hydrophobicity.** Bacterial cell surface hydrophobicity of LAB isolates was assessed by using bacterial adhesion to hydrocarbons (BATH) method according to the previous protocol [27]. The percent hydrophobicity (H%) was calculated as follows:

$$H\% = [(A_0 - A_1/A_0)] \times 100$$

Where, $A_0$ and $A_1$ are absorbance measured initial and after solvent extraction.

**Auto-aggregation.** The auto-aggregation ability of the LAB isolates was tested according to previous protocol described [28]. The auto-aggregation was measured as follows.

$$\% = [1 - (At/A0) \times 100]$$

Where, A0 is the absorbance at initial and At is absorbance at 24 h of incubation.

**β-galactosidase activity.** The β-galactosidase activity of the probiotic isolates was determined as previously described [29]. Active LAB cultures were streaked on MRS agar plates containing 60 µL X-gal (5-bromo-4-chloro-3-indolyl-b-D-galactopyranoside) (MB069, HiMedia, India) and 10 µL of IPTG (isopropyl-thio-b-D-galactopyranoside) (RM2578, HiMedia, India) and incubated at 37˚C for 24 h.

**Bile salt hydrolase activity.** Bile salt hydrolase (BSH) activity of the selected probiotic isolates were evaluated as previously described method [29]. Overnight grown LAB isolates were inoculated on MRS agar plates containing 0.5% (w/v) sodium taurodeoxycholic acid (TDCA-RM131, HiMedia, India) and plates were incubated at 37˚C for 72 h. The BSH activity was evaluated by formation of opaque halo of precipitate around the wells.

## Safety assessment of LAB isolates

**Hemolytic activity.** Hemolytic activity of selected probiotic isolates was evaluated according to the method described previously [30]. Overnight grown probiotic cultures were inoculated on Columbia blood agar (M144, HiMedia, India) plates supplemented with 5% (v/v) sheep blood, and plates were incubated at 37˚C for 48 h. Hemolytic activity was determined as α-hemolysis, β-hemolysis, and non-hemolysis based on the appearance of green zone, transparent zone and no halo around colonies, respectively.

**Antibiotic sensitivity.** Antibiotic sensitivity of selected LAB isolates was determined by disc diffusion method as previously reported [26]. Overnight grown LAB cultures were swabbed evenly on MRS agar plates using sterile cotton swabs and allowed to dry for 15 min. Different concentrations of antibiotic discs (HiMedia, India) were placed on test plates, and incubated at 37˚C for 48 h. After incubation, zone of inhibition (diameter in mm) around disc was measured using ruler. Results were presented as an average from triplicate and categorized as resistance R (≤15 mm), intermediate susceptibility I (16–20 mm) and susceptibility S (≥21 mm).

**Adhesion assay.** Adhesion of probiotic bacterial isolates to HT-29 intestinal cells was determined according to the method described [31, 32]. The human intestinal epithelial cell lines (HT-29) was procured from National Center for Cell Sciences (NCCS), Pune, India. The cell suspension with $1\times10^5$ cells prepared in 4 ml complete Dulbecco's modified eagle medium (DMEM, Thermo fischer scientific, USA) was transferred in six well tissue culture plates. The medium was changed every alternate day until cells attained full confluency. Before adherence assay, cells were fed with DMEM without antibiotics and washed thrice with PBS (pH 7.4). Fresh DMEM (without serum and antibiotics) was added to each well of the culture plates and incubated for 30 min at 37˚C. After incubation, $1\times10^6$ CFU/ml of probiotic bacterial isolates suspended in plain DMEM without antibiotics were added and incubated for 2 h at 37˚C with 5% $CO_2$. The cells were washed thrice with PBS and added 3 mL of methanol to each well for fixing the cells, followed by incubation at room temperature for 10 min. Later, the cells were stained with Giemsa stain (HiMedia, India) for 20 min at room temperature. The wells were washed with ethanol and plates were air-dried at room temperature. The number of probiotic bacteria present in 20 random microscopic fields was counted and grouped into non-adhesive, adhesive and strongly adhesive.

**Molecular based characterization of LAB isolates by 16S rRNA sequencing and MAL-DI-TOF MS.** The selected isolates were genotypically characterized based on 16S rRNA gene

sequencing which was carried out at Macrogen Inc., South Korea. The 16S rRNA partial sequences obtained were further analyzed by BLAST in the National Centre for Biotechnology Information (NCBI) database to check the similarity with existing sequences. The 16S rRNA gene sequences of the isolates were deposited in the GenBank database for accession numbers. The phylogenetic tree was constructed using Molecular Evolutionary Genetic Analysis (MEGA) software, version 7.0 with maximum likelihood method [33].

Bacterial identification using MALDI-TOF (VITEK MS) was carried out by direct spotting method. A single colony of LAB isolate was picked from pure culture plate using a disposable loop and directly spotted on a MALDI-TOF target slide and inserted into the mass spectrometer to record the fingerprints (BioMerieux, France; version V.3.2.0 software [34].

Characterization of LAB isolates using MALDI-TOF (VITEK-MS) was carried out by comparing with known bacterial information in the database. In peptide mass fingerprinting (PMF) matching, the MS spectrum of unknown bacteria was compared with the MS spectra of known bacteria of the database [35].

**Antagonistic activity of LAB isolates against bacterial and fungal pathogens.**  The antibacterial activity of LAB isolates against bacterial pathogens was determined by agar well diffusion method as previously reported [11]. Overnight grown LAB cultures were centrifuged at 10,000 g for 10 min and the supernatant was filtered using 0.22 μm syringe filters. Pathogenic bacterial suspensions were adjusted to 0.5 McFarland scale and spread on the surface of nutrient agar plates with sterile cotton swabs and 6 mm diameter wells were made using cork borer. A volume of 100 μl cell free supernatant was loaded in each well and incubated at 37°C for 48 h. After incubation, zone of inhibition (diameter in mm) was measured around the wells using ruler.

The antifungal activity of three LAB isolates was determined by agar overlay method [28]. Six fungal species like *A. flavus* MTCC 2788, *F. oxysporum*, *R. arrhizus* MTCC 24794, *Rhizopus* sp., and two *Mucor* sp. and two yeasts like *C. albicans*, *C. tropicalis* were selected to determine the antifungal activity of LAB isolates. Overnight grown LAB cultures were streaked on MRS agar plates, incubated at 37°C for 48 h. A volume of 50 μl of each fungal/yeast suspension ($10^5$ spores/ml) was evenly mixed with molten PDA/YPD soft agar (0.7% agar), overlaid on the LAB culture MRS plates and incubated at 28°C for 3–5 days. After incubation, the plates were examined for the inhibition zone around the streaked area of LAB isolates.

In both, the agar overlay method and well diffusion assay, zone of inhibition diameter >1mm was considered as positive inhibitory activity.

**Preparation of cell free supernatant (CFS).**  Extraction of antimicrobial compound from LAB-CFS was carried out according to the previously reported method [19, 36]. In brief, overnight grown LAB cultures (1%) were inoculated in 3 L of MRS broth and incubated at 37°C for 48 h. The cell free supernatant was collected by centrifugation at 10,000 rpm for 10 min, at 4°C and the supernatant was filtered through 0.45-μm filter membrane (Millipore). The supernatants were extracted using ethyl acetate (1:3 v/v) (109623, Merck, USA) and the solvent was concentrated in a rotary evaporator. The partial purified cell free supernatant was used for further analysis.

**Antifungal activity of LAB isolates against Covid-19 associated mucormycosis strains.** The antifungal activity of LAB-CFS was evaluated by the agar well diffusion method [37]. A volume of 20 μL of *Rhizopus* sp. and two *Mucor* sp. ($10^5$ spores /ml) suspensions were evenly mixed with PDA, poured into sterile petri plates and 6 mm diameter wells were made using sterile cork borer in the agar. The wells were loaded with 100 μl of 10-fold concentrated LAB-CFS and incubated for 5 days at 28°C and the inhibition zones were observed around the wells. The results were compared with *R. arrhizus* MTCC 24794.

**Detection and quantification of 3-PLA by HPLC and LC-MS.**  The characterization of 3-phenyllactic acid was carried out by HPLC and LC-MS according to the modified previous

method [36]. The preparative reversed phase HPLC (Shimadzu Nexera LC system) equipped with photodiode array detector (SPD-M20A) was used to detect and quantify PLA. The chromatographic separation was carried out using C18 column (250 mm × 4.6 mm, 5 μm particle size, Shimadzu). PLA was eluted under isocratic run using HPLC grade water and acetonitrile (100030, Merck, USA) (1:1) as a mobile phase. Both solvents were filtered through 0.22μm disposable filter membrane and degassed for 15 min before use. Cell free supernatants of LAB isolates were filtered through 0.22 μm syringe filters (Nylon, Whatman). The flow rate was set to 1.0 ml/min, the column temperature was maintained at 30˚C and a volume of 20 μl supernatant was injected manually. The elution was monitored using an ultraviolet (UV) wavelength set to 210 nm. Un-inoculated MRS broth was used as negative control and the 3-PLA (P7251, Sigma-Aldrich) was used as standard (1 mg/ml). Chromatographic data was analyzed with the software Lab Solutions version 6.72 (Shimadzu Corporation). Particular peak fraction was collected, incubated at 40˚C for removal of acetonitrile by evaporation and then freeze dried.

The purified fraction was analysed by LC-MS, performed on an ACQUITY and an UPLC system quadrupole ion trap mass spectrometer (Waters Corporation, Milford, MA) with electrospray ionization. For chromatographic separation, a BEH C18 column (4.6 mm × 150 mm, 5 μm; Waters) was used and temperature was maintained at 25˚C. The mobile phase was comprised of 0.1% formic acid in acetonitrile (A) and 0.1% formic acid in Milli Q water (B). The fraction was eluted with a linear gradient of solvent A increasing from 5% to 90% at a flow rate of 1.0 ml /min for 10 min. The HPLC output was split to a reduced flow rate of 0.2 ml/min before entering the electrospray ionization source. The mass scan range was m/z 100–500.

**Determination of minimum inhibitory concentration (MIC) of 3-phenyllactic acid.** The MIC of PLA against *R. arrhizus* MTCC 24794, *Rhizopus* sp. and two *Mucor* sp. were determined as previously described [38]. A volume of 20 μl of *Rhizopus* sp. and two *Mucor* sp. suspensions ($10^5$ spores/ml) were evenly mixed with molten PDA medium, poured into petri plates and 6 mm diameter wells were made using sterile cork borer in the agar plates. A volume of 100 μl 3-phenyllactic acid at different concentrations (5, 10, 15. . ..210 mg/ml) were loaded in the wells and incubated at 28˚C for 5 days. The MIC was considered when there was a visible growth inhibition after incubation at the lowest concentration of PLA.

**Microscopic observation of fungal growth inhibition.** The fungal growth inhibition assay was conducted in 6 well microtitre plates to assess the inhibitory activity of PLA on *Rhizopus* sp. and *Mucor* sp. A volume of 20 μl *Rhizopus* sp. and *Mucor* sp. spore suspensions ($10^5$ spores/ml) were added in potato dextrose broth separately and mixed with MIC of PLA which were considered as treated and the culture broth without PLA were considered as untreated. The treated and untreated suspensions were made up to 2 ml with sterile distilled water and incubated at 28˚C for 48–72 h. After incubation, samples were observed under EVOS live cell imaging inverted microscope and images were captured.

## Statistical analysis

All the experiments were carried out in triplicates and the results were calculated as mean ± standard deviation of triplicate. The results were statistically subjected to analysis of variance (ANOVA) using Tukey's post-hoc one-way analysis ($p < 0.05$).

## Results

### Screening and characterization of LAB isolates

A total of 129 strains were isolated from different sources. Among them, 19 strains were found to be Gram positive, catalase negative and non-spore forming, therefore considered as presumptive lactic acid bacteria. Finally, three strains were selected based on potential probiotic

**Table 1. Primary probiotic characteristics of lactic acid bacterial (LAB) strains.**

| Isolates | Autoaggregation (%) | Hydrophobicity (%) | | Temperature tolerance (˚C) | BSH activity | NaCl tolerance (%) |
|---|---|---|---|---|---|---|
| | | xylene | toluene | | | |
| BMOBR013 | 94.82±0.71 | 94.28±0.05 | 97.05 ± 0.06 | 20–50 | + | 4–9.5 |
| BMOBR061 | 94.77±0.86 | 96.45±0.96 | 96.75 ± 0.57 | 20–50 | + | 4–9.5 |
| BMOBR041 | 95.66±0.64 | 94.81±2.24 | 96.96 ± 0.57 | 20–50 | + | 4–9.5 |
| *L. acidophilus* | 94.33±0.03 | 95.51±1.37 | 94.99 ± 0.12 | 20–45 | - | 4–6.5 |

'+' indicates presence and '–' indicates absence of activity. Auto-aggregation and hydrophobicity were performed in triplicates, and data expressed as mean ± S.D (p <0.05).

properties including antimicrobial activity. The three bacterial strains thrived in temperature ranging from 20 to 50˚C and similarly, could tolerate up to 9.5% NaCl concentration (Table 1). Further, the selected strains were characterized by 16S rRNA sequencing and MALDI-TOF MS spectrometry. The 16S rRNA sequence of BMOBR013 isolate from human milk (long rod) and BMOBR061 isolate from buffalo milk (short rod) showed 100% similarity with *Lactobacillus pentosus*. Whereas BMOBR041 isolate from buffalo milk showed 99.9% similarity with *Pediococcus acidilactici*. The sequences of three isolates BMOBR013, BMOBR041 and BMOBR061 were submitted to NCBI GenBank database and acquired accession numbers MN880113, MN880127, and MN880129, respectively. A phylogenetic tree was constructed using MEGA software, version 7.0 by maximum likelihood method (Fig 1).

### *In vitro* assessment of probiotic attributes

Survival of probiotic LAB when taken orally and exposed to gastric juice and bile is a prerequisite step for reaching intestine, colonization and metabolic activity. In this study, isolates *L. pentosus* BMOBR013, *L. pentosus* BMOBR061, and *P. acidilactici* BMOBR041 showed the survival rate of ≥80% at pH 2.5 (1% pepsin) and ≥75% at 0.3% bile after 2 h of exposure with a significant variability ($p < 0.05$) (Fig 2a). The survival of the isolates BMOBR013, BMOBR061 and BMOBR041 at pH 2.5 containing 1% pepsin was found to be 98.8%, 97.1%, 90.2%, and

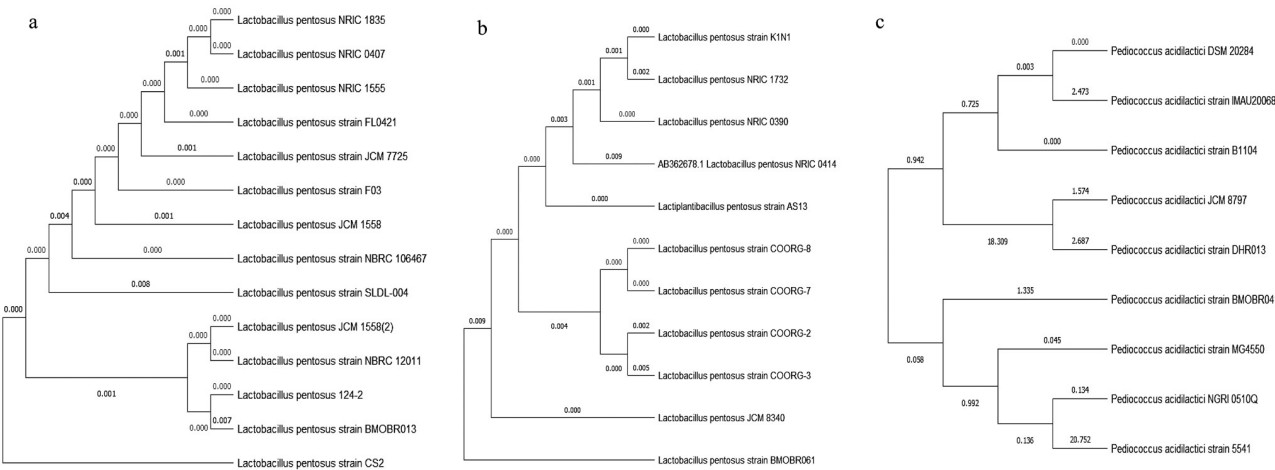

**Fig 1.** Phylogenetic analysis of the LAB isolates based on 16S rRNA gene sequence a) *Lactobacillus pentosus* BMOBRO13, b) *Lactobacillus pentosus* BMOBRO61 and c) *Pediococcus acidilactici* BMOBRO41. The dendrogram was constructed based on the BLAST algorithm and maximum likelihood method using MEGA X software with 1000 bootstrap replicates.

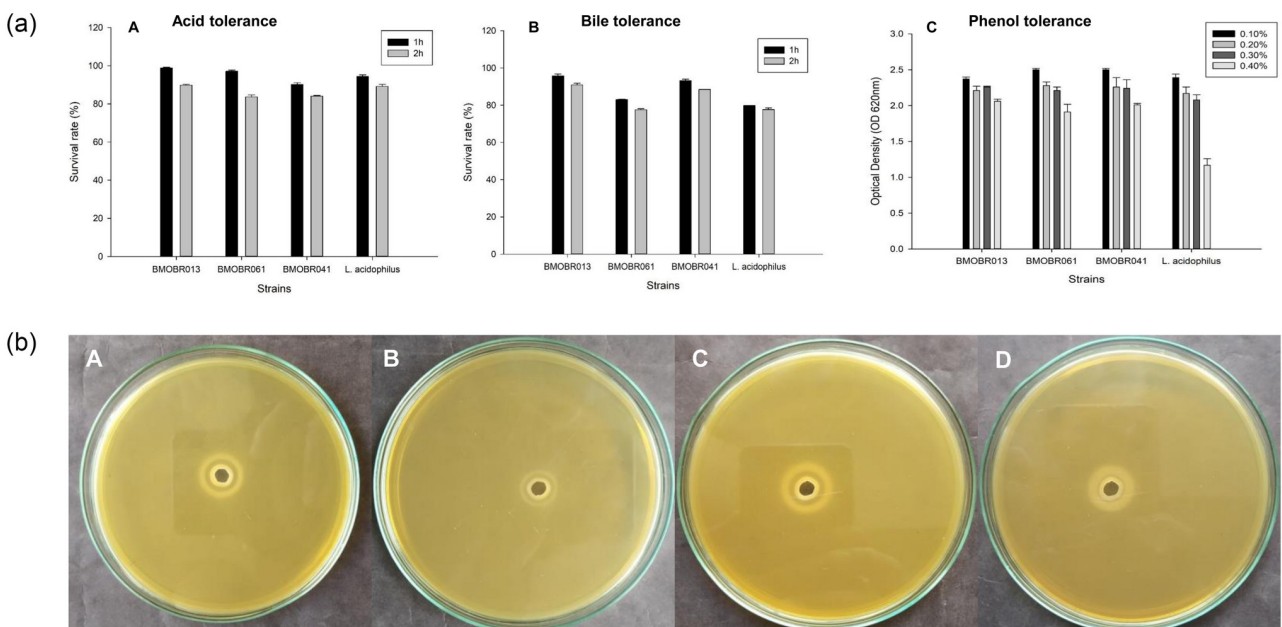

**Fig 2.** **a**. *In vitro* cell viability of bacterial isolates BMOBR013, BMOBR061 and BMOBR041 (A) acid tolerance, (B) bile tolerance (0.3%), and (C) phenol tolerance. All the three experiments carried out in triplicates and error bars indicate standard deviation of the mean values obtained from statistical analysis by Tukey's post-hoc analysis ($p < 0.05$). **b**. Bile salt hydrolase activity of LAB strains (A) *L. pentosus* BMOBR013, (B) *L. pentosus* BMOBR061, (C) *P. acidilactici* BMOBR041, (D) *L. acidophilus* NCIM 2285 on MRS-sodium taurodeoxycholic acid.

89.8%, 83.7%, 84.1%, after 1 and 2 h, respectively. The survival of the same isolates in bile was found to be 95.6%, 82.9%, 93.2%, and 90.9%, 77.5%, 88.4% after 1 and 2 h, respectively. Assessment of safety by hemolysis is an essential and one of the important properties of probiotic candidate. None of the strains showed hemolytic activity. Thus, the three isolates were selected for further probiotic characterization. The same LAB strains showed remarkable auto-aggregation ranging from 94.77 to 95.66% at 24 h and hydrophobicity ranging from 94.28 to 96.45%, and 96.75 to 97.05% for xylene and toluene, respectively (Table 1). The auto-aggregation and hydrophobicity of three LAB isolates differ significantly ($p < 0.05$). This result indicates that these strains are potentially capable of adhering to epithelial cells. The adhesion efficiency of three selected strains to HT-29 cells was examined by direct microscopic observation after Giemsa staining. The adhesion ability of isolates was strain-dependent. Three strains were strongly adhesive and the results showed significant count more than 100 bacteria attached to the HT-29 cells around the 20 microscopic fields. In qualitative screening of β-galactosidase activity, three strains showed the presence of blue colour colonies on culture plates, and these probiotic strains can reduce the lactose intolerance. The BSH activity of three strains showed the ability to hydrolyse sodium salt of taurodeoxycholic acid (TDCA), which is represented as opaque halo of precipitation around the wells (Fig 2b). The tolerance of LAB isolates to phenol concentrations (0.1–0.4%) was found to be efficient (Fig 2a). The results showed different degrees of tolerance to phenol and were capable of growing efficiently in the presence of 0.4% phenol with OD values >1.9. The viability of LAB isolates differs significantly ($p < 0.05$) with respect to phenol concentration.

## Antibiotic sensitivity

The antibiotic sensitivity of *L. pentosus* BMOBR013, *L. pentosus* BMOBR061 and *P. acidilactici* BMOBR041 isolates were assessed with 12 different antibiotics by disc diffusion method. All

**Table 2. Antibiotic sensitivity of three LAB isolates using disc diffusion method.**

| Antibiotics | Concentration (μg) | BMOBR013 | BMOBR061 | BMOBR041 | *L. acidophilus* |
|---|---|---|---|---|---|
| Chloramphenicol (C) | 30 | S | S | S | S |
| Tetracycline (TE) | 30 | S | S | S | I |
| Erythromycin (E) | 15 | I | S | S | S |
| Ampicillin (AMP) | 10 | S | S | I | S |
| Clindamycin (CD) | 2 | R | I | I | R |
| Ceftriaxone (CTR) | 10 | I | R | R | R |
| Kanamycin (K) | 30 | R | R | R | R |
| Lincomycin (L) | 10 | R | R | R | R |
| Streptomycin (S) | 10 | R | R | R | R |
| Rifampicin (RIF) | 5 | R | R | R | R |
| Ciprofloxacin (CIP) | 5 | R | R | R | R |
| Gentamycin (GEN) | 10 | R | R | R | R |
| Vancomycin (VA) | 30 | R | R | R | R |
| Penicillin-G (P) | 10 Units | R | R | R | R |

S = Sensitive; I = Intermediate; R = Resistant. The experiment was performed triplicate in each isolate

the three isolates showed resistant to vancomycin, kanamycin, streptomycin, gentamycin, lincomycin, rifampicin, ciprofloxacin, and penicillin-G but they were sensitive to chloramphenicol, erythromycin, ampicillin and tetracycline. The probiotic bacterium, *L. acidophilus* NCIM 2285 was used as referral probiotic strain (Table 2).

## Antimicrobial activity against bacterial and fungal pathogens

Antimicrobial activity is one of the significant properties of probiotics. In the current study, the probiotic strains were tested for antimicrobial activity against standard pathogenic bacteria. All the three LAB strains showed greater than 9 mm zone of inhibition against all the standard bacterial pathogens. The probiotic isolates *L. pentosus* BMOBR013 and *L. pentosus* BMOBR061 showed highest antimicrobial activity against *P. aeruginosa* with inhibition zones of 24.6 and 19.3 mm, respectively (Fig 3).

The antifungal activity of the LAB strains was tested against *A. flavus* MTCC 2788, *F. oxysporum*, *C. albicans*, *C. tropicalis*, *R. arrhizus* MTCC 24794, *Rhizopus* sp., and two *Mucor* sp. by the agar overlay method. The probiotic strains BMOBR013, BMOBR061, and BMOBR041 showed variable degrees of fungal inhibition after 4 days of incubation (Fig 4). Antifungal activity of three LAB-CFS against *R. arrhizus* MTCC 24794, *Rhizopus* sp. and two *Mucor* sp. using agar well diffusion method showed efficient inhibitory activity on par with standard PLA (Fig 5).

## Characterization and quantification of PLA

The antagonistic metabolite PLA produced by three LAB isolates was detected and quantified by HPLC, the retention time of the standard 3-PLA was found to be 7.7 which is similar to the PLA of LAB isolates (Fig 6). The isolate *L. pentosus* BMOBR013 has the ability to produce more PLA (0.441 g/L) than other two strains *P. acidilactici* BMOBR041 and *L. pentosus* BMOBR061 which produced 0.294 and 0.165 g/L, respectively. Further, confirmation and molar mass of PLA was carried out using LC-MS and illustrated in Fig 7. The molar mass of the LAB produced PLA was found to be 166 g/mol, which is in accordance with the standard

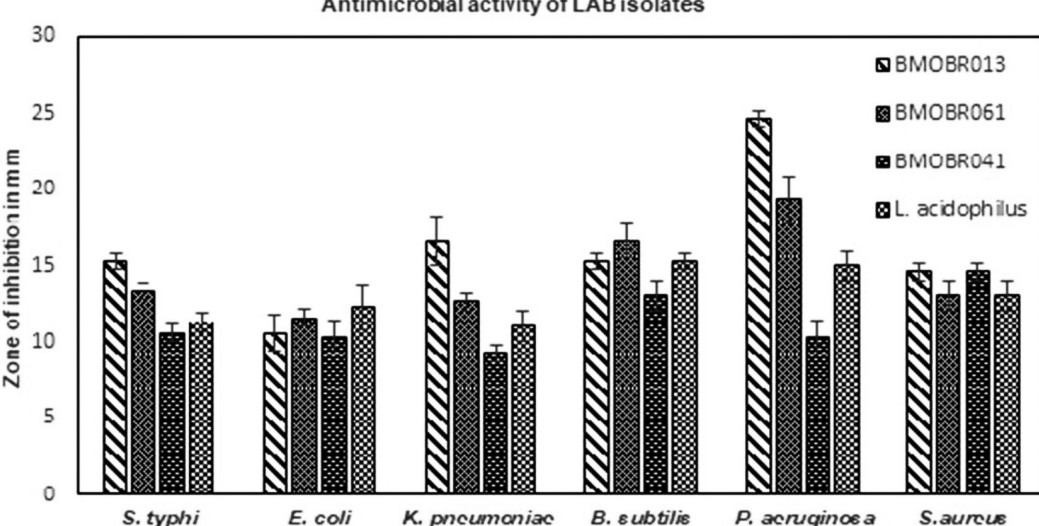

**Fig 3.** Antimicrobial activity of LAB isolates (A) *L. pentosus* BMOBR013, (B) *L. pentosus* BMOBR061, (C) *P. acidilactici* BMOBR041, (D) *L. acidophilus* NCIM 2285 against standard bacterial pathogens. Experiment was carried out in triplicate, and data expressed as the mean ± S.D. of three independent experiments.

PLA(Sigma-Aldrich). The minimum inhibitory concentration (MIC) of HPLC eluted PLA against *R. arrhizus* MTCC24794, *Rhizopus* sp., and two *Mucor* sp. was found to be 180 mg/ml and is illustrated in the Fig 8. The MIC of PLA on pathogenic fungal strains showed mycelial growth inhibition, whereas in untreated samples i.e., without PLA, there was no inhibition of mycelial growth was observed with both the species of fungal pathogens (Fig 9).

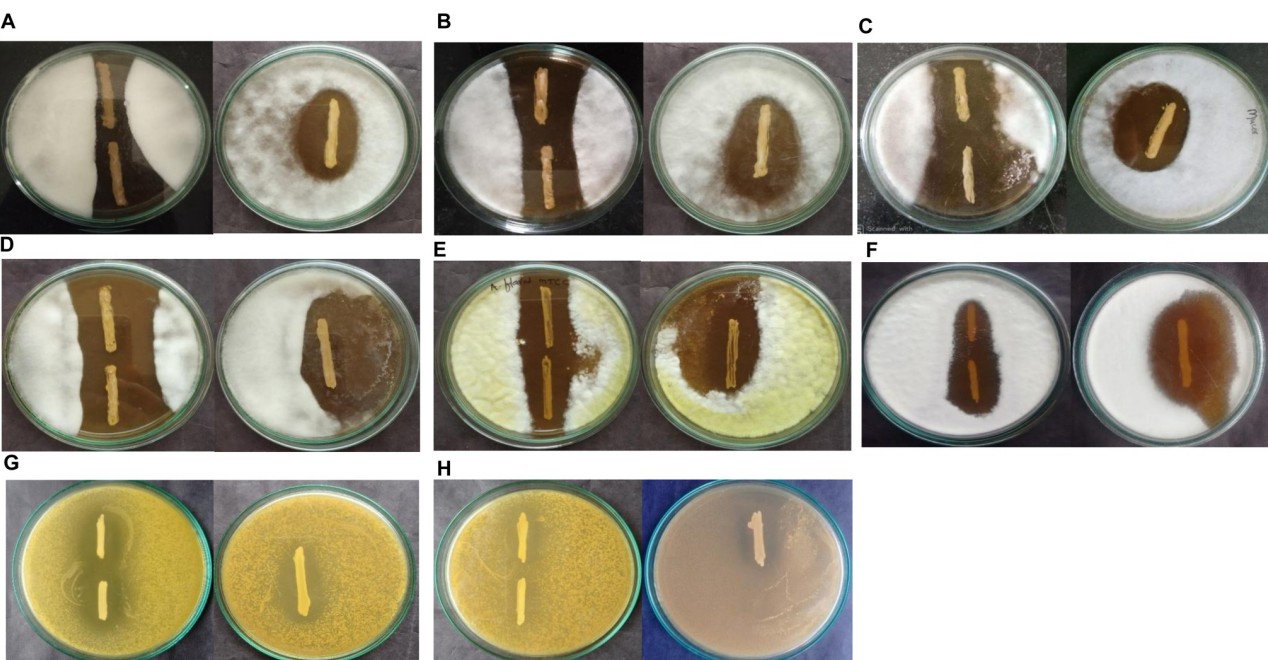

**Fig 4.** Antifungal activity of two *L. pentosus* strains and one *P. acidilactici* against fungal pathogens by agar over lay method (A) *Rhizopus arrhizus* MTCC 24794, (B) *Rhizopus* sp., (C) *Mucor* sp. 1, (D) *Mucor* sp. 2, (E) *Aspergillus flavus* MTCC 2788, (F) *Fusarium oxysporum*, (G) *Candida albicans*, (H) *Candida tropicalis*.

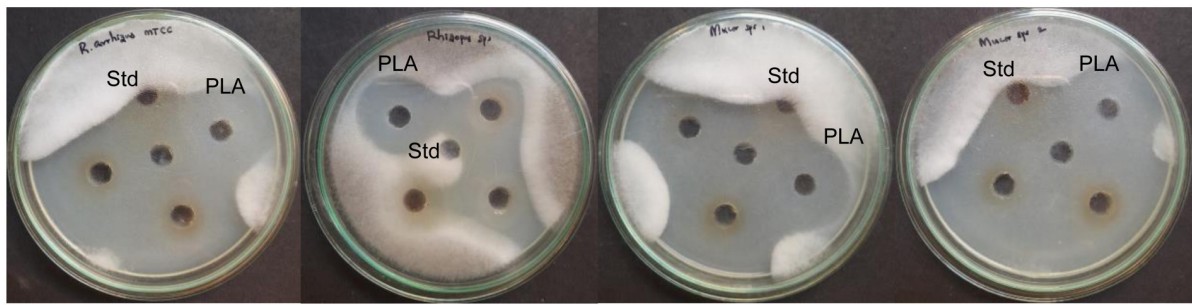

**Fig 5. Antifungal activity by the agar well diffusion method showing inhibition of *Rhizopus arrhizus* MTCC 24794, *Rhizopus* sp., *Mucor* sp. 1, and *Mucor* sp. 2 with cell free supernatant of three LAB isolates on par with standard strain (*L. acidophilus*) and 3-PLA (Sigma Aldrich).**

## Discussion

In the present study, three LAB strains were isolated from human breast milk and buffalo milk which were characterized for their potential probiotic properties. The selected isolates *L.*

**Fig 6.** HPLC profile of 3-phenyllactic acid produced by (A) *L. pentosus* BMOBR013, (B) *L. pentosus* BMOBR061, (C) *P. acidilactici* BMOBR041, (D) *L. acidophilus* NCIM 2285, (E) 3-PLA (Sigma Aldrich).

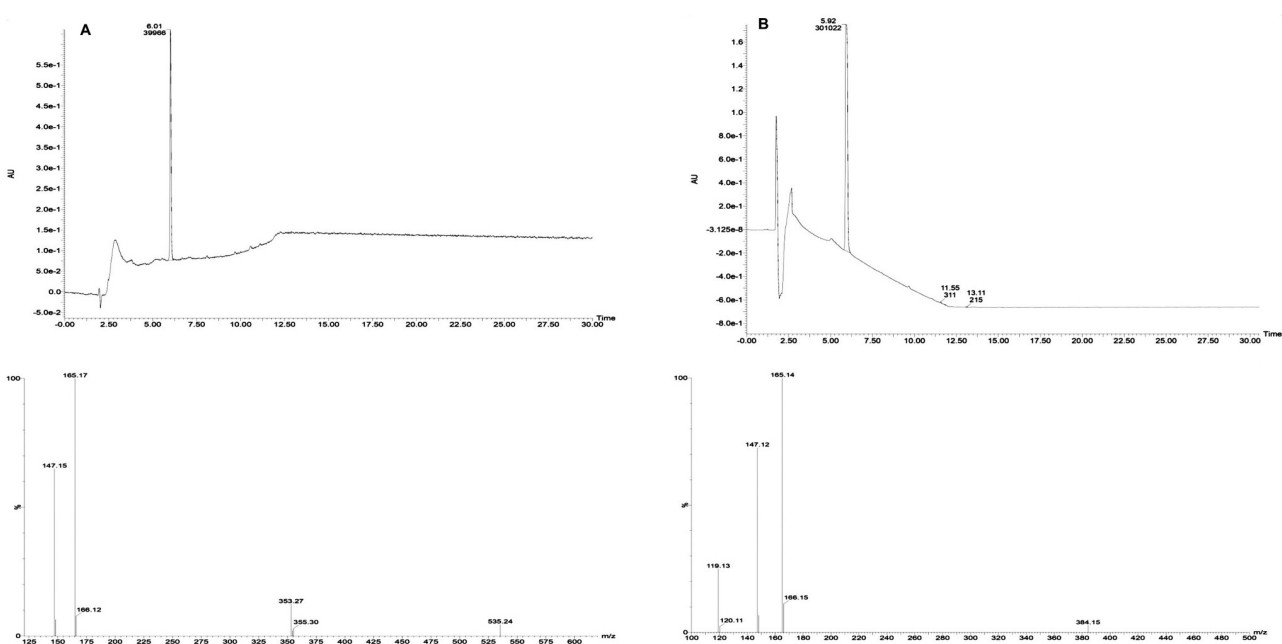

**Fig 7.** Confirmation and molar mass detection of 3-phenyllactic acid by LC-MS (A) Standard 3-PLA (Sigma Aldrich), (b) LAB isolates.

*pentosus* (BMOBR013, BMOBR61), and *P. acidilactici* BMOBR041 showed more than 83% survival in simulated gastric juice (1% pepsin, pH 2.5) and more than 77% survival at 0.3% ox bile after 2 h of incubation in the medium. The present findings are higher in accordance with previous studies [13, 39]. The bacterial isolate *L. pentosus* BMOBR013 showed maximum survival of 90% in the gastrointestinal conditions compared to *P. acidilactici* BMOBR041 and *L. pentosus* BMOBR061. In the present study, the survival of probiotic isolates in low pH and bile, significantly vary from one strain to another. The BSH activity of probiotic bacteria determines its tolerance to bile salts and reduction of serum cholesterol in the host [40]. The cholesterol-lowering effect of probiotic bacteria is primarily based on BSH activity [41]. Our findings are similar to previous studies [42] who reported bile salt precipitation around the colonies of LAB. The growth of three LAB isolates was observed at temperatures ranging from 20–50°C and able to tolerate 4–9.5% NaCl concentrations while *L. acidophilus* NCIM 2285 could tolerate up to 45°C and 6.5% NaCl. These findings are in agreement with previous studies [26, 43] where they reported the tolerance of LAB to higher temperature and salt concentration.

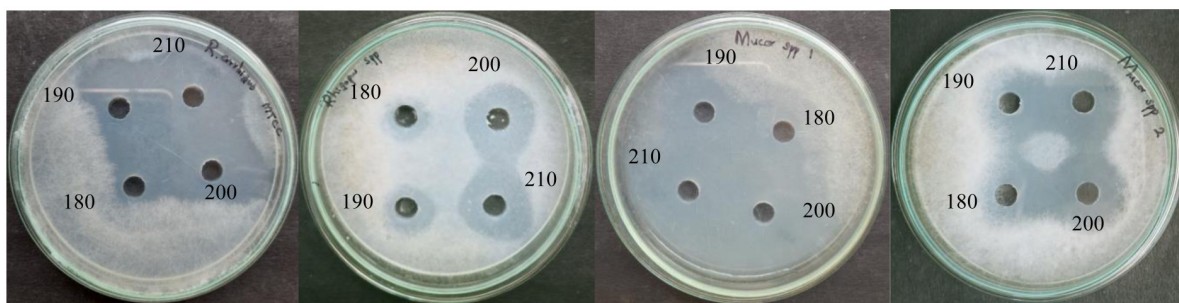

**Fig 8. Evaluation of minimum inhibitory concentration of 3-phenyllactic acid against *Rhizopus arrhizus* MTCC 24794, *Rhizopus* sp., *Mucor* sp. 1, and *Mucor* sp. 2.**

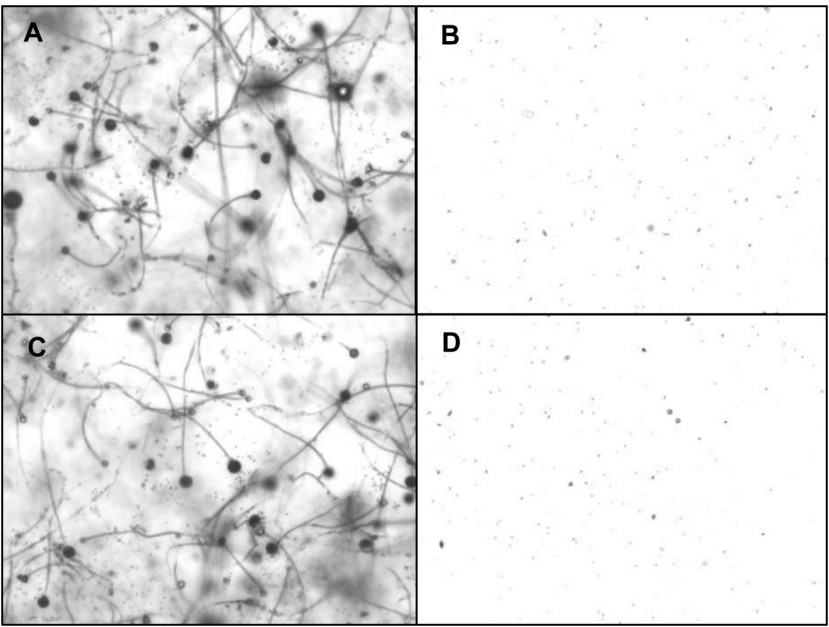

**Fig 9. Mycelial inhibition of *Rhizopus* sp. and *Mucor* sp. using HPLC eluted 3-phenyllactic acid under EVOS live cell imaging microscopy resulted in minimum inhibitory concentration (180 mg/ml).** (A) *Rhizopus* sp. (untreated), (B) *Rhizopus* sp. treated with PLA, (C) *Mucor* sp. (untreated), (D) *Mucor* sp. treated with PLA.

The absence of haemolytic activity is to be evaluated as a safety aspect of the probiotic candidate. All the three LAB isolates showed γ-hemolytic activity (i.e., no hemolysis) indicating non-virulence and lack of hemolysin. The antibiotic sensitivity of all the three strains revealed resistant to vancomycin, gentamicin, streptomycin, kanamycin, ciprofloxacin, lincomycin, rifampicin, and penicillin. The present findings are similar to previous studies [25, 44] where they reported similar resistance of LAB with the same antibiotics. Strains that can be used as probiotics should show resistance to some antibiotics to survive in the GIT and to impart beneficial activities to the host [26]. The intrinsic antibiotic resistant nature of probiotic bacteria can be administered along with antibiotics use in the treatment of intestinal disorders or infections, and it could be useful for restoring the gut microbiota after antibiotic treatment [26].

The surface properties like auto-aggregation and hydrophobicity are key factors that determine the adhesion and colonization efficiency of probiotic bacteria to intestinal epithelial cells, which would enable biofilm formation to protect the host from pathogenic colonization [26, 27]. In the present study, three LAB strains showed variable hydrophobicity ranging from 94 to 97%, which is in agreement with Fonseca *et al* [45] who reported 96.06% hydrophobicity by *L. plantarum*. In comparison to the results attained by Yasmin *et al* [27] and Fonseca *et al* [45], our probiotic strains showed a higher auto-aggregation value of 95.6%. Similar to present findings, Tuo *et al* [46] reported that the probiotic strain *L. plantarum* showing higher auto-aggregation also showed high hydrophobic activity. Therefore, high hydrophobicity and auto-aggregation of probiotic bacteria are found to be positively correlated with strong adhesion ability [47]. A similar result was observed in the current study, where the LAB isolates showed strong adhesion towards intestinal cancer cell lines (HT-29) indicating that colonizing ability to the intestinal epithelium that represents a barrier to protect the GIT from pathogens.

Lactose maldigestion and/or intolerance are due to lack of β-galactosidase enzyme, lactose indigestion can be improved by consuming probiotic bacteria that produce lactose hydrolyzing enzyme i.e., β-galactosidase. Our results are consistent with the previous study [29].

Hence, LAB isolates could be beneficial in the food industry and also play an effective role in the treatment of lactose intolerance as nutritional supplement. The resistance to phenol is an important factor for probiotic bacteria to survive in the GIT, where the intestinal microbiota can deaminate few aromatic acids that are derived from endogenous or dietary proteins and may lead to the formation of phenols [28]. In the present study, three probiotic strains are found to be tolerant to 0.4% phenol with OD value of >1.9. These findings are in agreement with Reuben *et al* [26] who reported <0.3 OD value with 0.4% phenol tolerance.

Currently, several molecular tools are used for the characterization of LAB species, particularly 16S rRNA gene sequencing, polymerase chain reaction (PCR), and its related PCR-based techniques. In recent years, Matrix-assisted laser desorption/ionization-time of flight mass spectrometry (MALDI-TOF MS) has been introduced with marked success into routine clinical microbiological laboratories for bacterial identification [48]. The identification of LAB species success rate was higher using MALDI-TOF MS i.e., 93% specificity at the species level in comparison with a molecular method such as 16S rDNA gene sequencing (77%) [49]. In the present study, the results showed 100% congruency between 16S rRNA sequence analysis and MALDI-TOF MS in the identification of two *L. pentosus*, and one *P. acidilactici*. The characterization of LAB species based on MALDI-TOF MS could be an alternative to biochemical and molecular biology-based identification methods, due to its rapid, accurate, ease of use and low cost per analysis, high throughput use, sensitivity, and specificity for identification of LAB species [49, 50].

Probiotic bacteria produce antimicrobial compounds and it is one of the crucial characteristics for competitive exclusion of pathogens. In this study, *L. pentosus* (BMOBR013 and BMOBR061) and *P. acidilactici* BMOBR041 showed strong inhibitory activity against gram-positive and gram-negative bacteria including *B. subtilis*, *S. aureus*, *K. pneumoniae*, *E. coli*, *P. aeruginosa*, and *S. typhi*. Also, showed inhibition against *A. flavus* MTCC 2788, *F. oxysporum*, *C. albicans*, and *C. tropicalis*. Our results are in accordance with previous studies [43, 51–53] that studied on antimicrobial activity of probiotic LAB strains. Among the three isolates, *L. pentosus* BMOBR013 showed highest antibacterial activity against *P. aeruginosa*. In the present study, the three strains showed a broad range of antimicrobial activity against bacteria, yeasts and molds. The results of our study differ with Okkers *et al* [54] who reported the inhibition of *C. albicans* by *L. pentosus*, but not towards filamentous fungi. The antibacterial activity of PLA was first reported in *Geotrichum candidum*, and it was also found that D-PLA exerts stronger antibacterial activity than L-PLA [55]. The production of PLA was first characterized by Lavermicocca *et al* [56], who purified the antagonistic compound from cell-free supernatant of *L. plantarum* 21B. Bacteriocin like nisin produced by LAB whose inhibitory effects restricted to gram-positive bacteria whereas 3-PLA possesses a broad-spectrum inhibitory activity against bacteria and fungi [57]. The use of PLA is limited due to its low production by the known LAB species, for large-scale production, high yielding strains are required. The detection and quantification of antimicrobial compound PLA produced by our LAB isolates were carried out using HPLC and compared with the standard DL-3-PLA (Sigma-Aldrich). The HPLC profile of standard PLA has a similar retention time (7.7) with the PLA of our isolates BMOBR013, BMOBR061, and BMOBR041. Further, identification and molar mass of PLA was confirmed by LC-MS analysis. Zhang *et al* [24] reported that *L. plantarum* IMAU10124 showed 0.229 g/L PLA which is less than the PLA produced by our isolate *L. acidophilus* BMOBR013 in MRS medium. Compared to previous studies, *L. pentosus* BMOBR013 is found to be the highest PLA (0.441 g/L) producing probiotic strain followed by *P. acidilactici* BMOBR041 (0.294 g/L) without optimizing the production parameters.

The pandemic of Covid-19 with its first and second waves has driven the experiments to study LAB strains against Covid-19 associated mucormycosis causing fungal organisms. Among all the mucormycosis cases, rhino-ortal-cerebral infection is the most commonly

encountered form, followed by the pulmonary and cutaneous type [1, 5]. The pulmonary mucormycosis is seen in patients with hematological malignancies, which invade lung-adjacent organs i.e., pericardium, mediastinum, and chest wall with an overall mortality rate of 76% [5]. Patients who are admitted and exposed to mechanical ventilation, various emergency procedures, breaches in asepsis, and prolonged hospitalization may have chances of getting opportunistic secondary bacterial and fungal infections [16]. The prolonged use of steroids has often been linked with numerous opportunistic fungal infections such as mucormycosis and aspergillosis, even a short-term steroid treatment has been recently reported to be associated with mucormycosis mainly in diabetes mellitus patients [1]. The excessive use of steroidal drugs to control Covid-19 may be directly associated with the suppressing immunity of the patients; consequently, it allows opportunistic fungal colonization, leading to mucormycosis at any phase of the disease [58, 59]. In India, the major drawback in the management of mucormycosis infections is lacuna in treatment procedures and the financial instability of patients to afford liposomal AmB drug [8]. The administration of high doses of steroids used in treating mucormycosis leads to severe side effects imparting a major impact on the kidney [60]. Several studies have shown that probiotics used as a nutritional supplement may play a beneficial role in maintaining immune homeostasis during Covid-19 associated mucormycosis, and decrease the severity of GIT infections, candidiasis, and other fungal infections [17]. Probiotics are used as nutritional supplements in Covid-19 patients to diminish opportunistic secondary infections, they could be crucial in maintaining an optimal immune system and with their ability to modulate gut microbiota [16]. Thus, the administration of probiotics can be a significant way to resolve the severity of fungal infections. While many researchers have reported the antifungal activity of probiotic bacteria, the exact underlying mechanism is poorly understood. There is no validated specific laboratory biomarker for the diagnosis of mucormycosis co-infection incidence in Covid-19 unlike (1, 3)-β-D-glucan and galactomannan markers which are used for diagnosis of other invasive fungal infections [61]. The results obtained by our probiotic LAB-CFS showed good inhibitory activity against *R. arrhizus* MTCC 24794, *Rhizopus* sp. and two *Mucor* sp. Our study on MIC of PLA against mucormycosis causing fungal strains like *Rhizopus* sp. and 2 *Mucor* sp. showed 180 mg/ml. The present findings of MIC are on par with previous studies where they reported the range of inhibitory concentration of 50–166 mg/ml of PLA against fungal pathogens like *A. niger*, *A. flavus*, and *Rhizopus stolonifer* [56]. The live-cell imaging studies showed that MIC of PLA could inhibit the mycelial growth of mucormycosis causing *Rhizopus* sp. and *Mucor* sp. To the best of our knowledge, this is the first report of antifungal activity of LAB against Covid-19 associated mucormycosis infection. Further, there is a need to understand the mechanism of action of antifungal compound against mucormycosis infection. The present work gives a primary insight into the role of probiotics in fatal infections like mucormycosis and their effects paving the way in the treatment of infection.

## Conclusion

The LAB isolates of our study like *L. pentosus* (BMOBR013 and BMOBR061) and *P. acidilactici* BMOBR041, isolated from human and buffalo milk showed potential probiotic characteristics along with efficient antagonistic activity against bacterial and fungal pathogens including mucormycosis causing strains. Further studies proved the other health benefits of the three isolates. The PLA produced from LAB is commonly used as a bio preservative in the food industry as it extends the shelf life of food and food stuff, also inhibiting the mucormycosis infection causing *Rhizopus* sp. and *Mucor* sp. More insights into exact mechanism of antagonistic compound PLA against mucormycosis causing fungi especially *Rhizopus* sp. and *Mucor* sp. would create ways for effective prophylaxis of the disease.

## Author Contributions

**Conceptualization:** Bhima Bhukya.

**Data curation:** Ravikumar Lunavath, Saddam Hussain Mohammad.

**Formal analysis:** Ravikumar Lunavath, Saddam Hussain Mohammad, Kiran Kumar Bhukya, Anuradha Barigela.

**Funding acquisition:** Bhima Bhukya.

**Investigation:** Bhima Bhukya.

**Methodology:** Ravikumar Lunavath, Kiran Kumar Bhukya, Anuradha Barigela.

**Resources:** Anil Kumar Banothu, Bhima Bhukya.

**Software:** Chandrasekhar Banoth.

**Supervision:** Bhima Bhukya.

**Validation:** Ravikumar Lunavath, Chandrasekhar Banoth.

**Visualization:** Ravikumar Lunavath, Kiran Kumar Bhukya, Anuradha Barigela, Anil Kumar Banothu.

**Writing – original draft:** Ravikumar Lunavath, Saddam Hussain Mohammad.

**Writing – review & editing:** Anil Kumar Banothu, Bhima Bhukya.

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
