## [Decision Letter · Decision Letter 0]

4 Nov 2022

PONE-D-22-20891Antimycotic effect of 3-phenyllactic acid produced by probiotic bacterial isolates against Covid-19 associated mucormycosis causing fungiPLOS ONE

Dear Dr. Bhukya,

Thank you for submitting your manuscript to PLOS ONE. After careful consideration, we feel that it has merit but does not fully meet PLOS ONE’s publication criteria as it currently stands. Therefore, we invite you to submit a revised version of the manuscript that addresses the points raised during the review process.

Write abbreviations in full when appeared 1^st^ time (e.g., PLA).Avoid using keywords which are used in the title of the manuscript.Authors should carefully proofread their article for English (e.g., line 116).Check last sentence of introduction (line 182).GenBank database for accession→ GenBank database for accession numbers (line 300).In Figure 1 - Can authors give the reason why Lactobacillus pentosus (TEZU174) and Lactobacillus pentosus (BMOBR013) did not join the clade with other Lactobacillus species?Figure resolutions should be improved.==============================

We look forward to receiving your revised manuscript.

Kind regards,

Ghulam Mustafa, PhD

Academic Editor

PLOS ONE

Journal Requirements:

2. Please provide additional details regarding participant consent from the volunteer lactating mothers. In the ethics statement in the Methods and online submission information, please ensure that you have specified (1) whether consent was informed and (2) what type you obtained (for instance, written or verbal, and if verbal, how it was documented and witnessed). If the need for consent was waived by the ethics committee, please include this information

“No”

“No authors have competing interests”

6. Please upload a new copy of Figures 2a and 6 as the detail is not clear. Please follow the link for more information: " ext-link-type="uri" xlink:type="simple">https://blogs.plos.org/plos/2019/06/looking-good-tips-for-creating-your-plos-figures-graphics/"
https://blogs.plos.org/plos/2019/06/looking-good-tips-for-creating-your-plos-figures-graphics/

Reviewers' comments:

Reviewer's Responses to Questions

**Comments to the Author**

1. Is the manuscript technically sound, and do the data support the conclusions?

Reviewer #1: Yes

Reviewer #2: Yes

2. Has the statistical analysis been performed appropriately and rigorously? 

Reviewer #1: Yes

Reviewer #2: Yes

3. Have the authors made all data underlying the findings in their manuscript fully available?

Reviewer #1: Yes

Reviewer #2: Yes

4. Is the manuscript presented in an intelligible fashion and written in standard English?

Reviewer #1: Yes

Reviewer #2: Yes

5. Review Comments to the Author

Reviewer #1: In this paper author report isolation and characterization of 3 lactic acid bacterial strains from human and buffalo milk. They performed various biochemical and molecular experiments to characterize the strains in detail, particularly those relevant to probiotic application of the strains. Based on these finding these 3 strains were selected as potential candidates for probiotic use. To further enhance the significance they also tested the isolated strains for their antifungal activity against Covid-19 associated fungal infection showing good inhibitory activity. Finally they characterized the bioactive component, identified as PLA and quantified it. These isolates produced good yields of PLA indicating their potential for producing this agent for commercial applications with process optimization. Overall the study reports interesting findings with great potential for application in food and health industry. Manuscript is well written, all experiments are performed using standard methods, data is also very well presented.

Following few minor changes may be incorporated for acceptance..

1. Seems introduction is too lengthy and more focused on Covid-19 rather than on LAB and how they can be used in health improvement, it should be appropriately revised to reflect the contents of this article. Some of this information can be moved to discussion also.

2. Ln209 - Clarify what sample was used to isolated Covid-19 associated fungi?

3. Ln252,253, few other places - "beta" is missing. Please correct.

4. Include cat# for all bacterial strains, cell line, media, reagents used. if available.

5. Table 2. data can be rearranged to sort antibiotics by Sensitive intermediate resistant ones.

6. Ln 469-498 can have its subheading for clarity. eg "Characterization quantification of PLA"

7. Figures resolution is poor. Labels are not visible.

8. Include statistics (p-values) where ever applicable.

Reviewer #2: The paper may be accepted for publication after incorporating following queries.

1. Authors may explain the reason for selecting specific 04 fungal strains to find the antimycotic studies i.e. Aspergillus flavus, Fusarium oxysporum, Candida albicans and Candida tropicalis? Why not other fungal strains to answer mucormycosis?

2. Authors can provide clear bar graphs for the figure 2a In-vitro cell viability?

3. Figure 2b, zone of inhibition in the agar plates was clearly seen can authors provide it?

4. HPLC profile in Figure 6 which was presented in the manuscript is not clear and it is recommended to provide better images for readers.

5. Figure 7, is unclear and it is recommended to provide clear confirmation and molar mass detection by LC-MS.

6. PLOS authors have the option to publish the peer review history of their article (what does this mean?). If published, this will include your full peer review and any attached files.

Reviewer #1: No

Reviewer #2: No

---

## [Author Response · Author response to Decision Letter 0]

27 Nov 2022

Response to the editor and reviewer comments

All the comments pointed out by the editor and the reviewers are carefully addressed in the revised manuscript. The deleted text is kept red in colour and added text is blue throughout the manuscript (marked).

Comments from the Editor

1. Write abbreviations in full when appeared 1st time (e.g., PLA).

Authors’ response: Thank you for your comment. As you suggested, now we have included expansions of abbreviations when they appear first time. 

2. Avoid using keywords which are used in the title of the manuscript. 

Authors’ response: Thank you for your advice. According to your suggestion, now we have incorporated new keywords in the revised manuscript. 

3. Authors should carefully proofread their article for English (e.g., line 116).

Authors’ response: Thank you for your comment. As per your suggestion, the manuscript is carefully proofread again and necessary corrections have been made. The additions are shown as blue text and deletions are in red text throughout the ‘marked manuscript’.

4. Check last sentence of introduction (line 182). 

Authors’ response: Thank you for your comment. Now our revised manuscript edited and proofread by a native English speaker.

5. GenBank database for accession→ GenBank database for accession numbers (line 300).

Authors’ response: The necessary changes are made as “GenBank database for accession numbers” in line 297 in the revised manuscript. 

6. In Figure 1 - Can authors give the reason why Lactobacillus pentosus (TEZU174) and Lactobacillus pentosus (BMOBR013) did not join the clade with other Lactobacillus species?

Authors’ response: Thank you for your comment. The error was occurred in the preparation of phylogenetic tree in the manuscript. Now we have constructed a new phylogenetic tree in the revised manuscript and replaced the old figure. 

7. Figure resolutions should be improved. 

Authors’ response: The resolution and quality of all the figures are improved and now in accordance with the journal standards.

Journal Requirements:

1. Please ensure that your manuscript meets PLOS ONE's style requirements, including those for file naming. The PLOS ONE style templates can be found at https://journals.plos.org/plosone/s/file?id=wjVg/PLOSOne_formatting_sample_main_body.pdf

and https://journals.plos.org/plosone/s/file?id=ba62/PLOSOne_formatting_sample_title_authors affiliations.pdf

Authors’ response: Thank you for your comments. We have corrected, formatted and edited the entire manuscript as per the journal’s guidelines.

2. Please provide additional details regarding participant consent from the volunteer lactating mothers. In the ethics statement in the Methods and online submission information, please ensure that you have specified (1) whether consent was informed and (2) what type you obtained (for instance, written or verbal, and if verbal, how it was documented and witnessed). If the need for consent was waived by the ethics committee, please include this information.

Authors’ response: The volunteer lactating mothers were informed about the experiment and taken a written consent from them for obtaining ethical approval. The same statement is included in the methods section of the manuscript and online submission portal.

Authors’ response: Thank you for the comment. We will ensure that the correct numbers are provided during the submission.

“No”

Authors’ response: The required changes have been made. 

The following is included in the ‘Funding’ information file as well as in the online portal.

Funding: This work was financially supported by the University Grants Commission (UGC), Govt. of India through CSIR-UGC fellowship (Ref. No: 21/06/2015(I)EU-V), the Ministry of Human Resource Development, Govt. of India through RUSA 2.0 program and Department of Science and Technology, Govt. of India through DST-PURSE program. The funders had no role in study design, data collection and analysis, decision to publish, or preparation of the manuscript.

Authors’ response: The funders had no role in study design, data collection and analysis, decision to publish, or preparation of the manuscript. The same statement is included in the ‘Funding’ information file.

Authors’ response: No authors received salary from funders. 

Authors’ response: NA

Authors’ response: The required changes have been included in the ‘Funding’ information file as well as in the online portal. 

“No authors have competing interests”

Authors’ response: ‘No authors have competing interests’ statement has been included in the cover letter as well as in the online portal. 

6. Please upload a new copy of Figures 2a and 6 as the detail is not clear. Please follow the link for more information: https://blogs.plos.org/plos/2019/06/looking-good-tips-for-creating-your-plos-figures-graphics/" https://blogs.plos.org/plos/2019/06/looking-good-tips-for-creating-your-plos-figures-graphics/

Authors’ response: Thank you for your kind suggestion. Now the revised manuscript is included 

with new copy of Figures 2a and 6. 

Authors’ response: The typographical error occurred in the preparation of 5th reference in the manuscript. Now we made a correction in the revised manuscript. 

Reviewer Comments to the Author

Reviewer #1

In this paper author report isolation and characterization of 3 lactic acid bacterial strains from human and buffalo milk. They performed various biochemical and molecular experiments to characterize the strains in detail, particularly those relevant to probiotic application of the strains. Based on these finding these 3 strains were selected as potential candidates for probiotic use. To further enhance the significance they also tested the isolated strains for their antifungal activity against Covid-19 associated fungal infection showing good inhibitory activity. Finally they characterized the bioactive component, identified as PLA and quantified it. These isolates produced good yields of PLA indicating their potential for producing this agent for commercial applications with process optimization. Overall the study reports interesting findings with great potential for application in food and health industry. Manuscript is well written, all experiments are performed using standard methods, data is also very well presented. 

Following few minor changes may be incorporated for acceptance. 

1. Seems introduction is too lengthy and more focused on Covid-19 rather than on LAB and how they can be used in health improvement, it should be appropriately revised to reflect the contents of this article. Some of this information can be moved to discussion also.

Authors’ response: Thank you for your valuable inputs. According to your suggestion, some introduction part of Covid-19 relevant information moved to discussion section in the revised manuscript. The effects of probiotics and its health benefits on Covid-19 associated co-infections information included in the revised manuscript. 

2. Ln209 - Clarify what sample was used to isolated Covid-19 associated fungi?

Authors’ response: Thank you for your comment. The Covid-19 associated fungal organisms like Rhizopus sp. and Mucor sp. have been isolated from nasal sinuses of mucormycosis affected patients. The same information has been included in materials and methods section of revised manuscript. 

3. Ln252,253, few other places - "beta" is missing. Please correct.

Authors’ response: Now the revised manuscript is included with beta (β) in Ln252, 253 and few other places in the revised manuscript.

4. Include cat# for all bacterial strains, cell line, media, reagents used. if available. 

Authors’ response: Thank you for your comment. Now we have included catalogue numbers wherever possible in the revised manuscript. 

5. Table 2. data can be rearranged to sort antibiotics by Sensitive intermediate resistant ones.

Authors’ response: Thank you for your comment. The Table 2. data has been rearranged like sensitive intermediate resistant in the revised manuscript.

6. Ln 469-498 can have its subheading for clarity. eg "Characterization quantification of PLA"

Authors’ response: Thank you for your comment. According to your suggestion, now we made "Characterization and quantification of PLA" as subheading in the revised manuscript.

7. Figures resolution is poor. Labels are not visible.

Authors’ response: Thank you for your suggestion. Now the revised manuscript is included with better resolution of all Figures. 

8. Include statistics (p-values) where ever applicable. 

Authors’ response: The revised manuscript is included with statistics (p-values) wherever applicable. 

Reviewer #2:

The paper may be accepted for publication after incorporating following queries.

1. Authors may explain the reason for selecting specific 04 fungal strains to find the antimycotic studies i.e. Aspergillus flavus, Fusarium oxysporum, Candida albicans and Candida tropicalis? Why not other fungal strains to answer mucormycosis?

Authors’ response: Thank you for your comment. There has been a significant increase in the incidence of these fungal infections in human and all these fungi are highly virulent in nature. Therefore, we have selected and worked on these species. 

2. Authors can provide clear bar graphs for the figure 2a In-vitro cell viability?

Authors’ response: Thank you for your comment. Now the revised manuscript is included with revised Figure 2a showing clear bar graphs. 

3. Figure 2b, zone of inhibition in the agar plates was clearly seen can authors provide it?

Authors’ response: Thank you for your comment. Some LAB strains produce bile salt hydrolase enzyme, which hydrolyses conjugated bile acids to release de-conjugated bile acids and amino acids. Bile acid precipitates around the wells (opaque halo) or the formation of opaque granular white colonies were considered to be indicate bile salt hydrolase activity. Isolated three LAB strains showed opaque halo of precipitation around the wells. Now the revised manuscript is provided with clear image of Figure 2b.

4. HPLC profile in Figure 6 which was presented in the manuscript is not clear and it is recommended to provide better images for readers.

Authors’ response: Thank you for your comment. Now the revised manuscript is provided with clear image of HPLC profile in Figure 6. 

5. Figure 7, is unclear and it is recommended to provide clear confirmation and molar mass detection by LC-MS.

Authors’ response: The All the figures are revised to improve the quality.

---

## [Editor Report · Decision Letter 1]

1 Dec 2022

Antimycotic effect of 3-phenyllactic acid produced by probiotic bacterial isolates against Covid-19 associated mucormycosis causing fungi

PONE-D-22-20891R1

Dear Dr. Bhukya,

We’re pleased to inform you that your manuscript has been judged scientifically suitable for publication and will be formally accepted for publication once it meets all outstanding technical requirements.

Kind regards,

Ghulam Mustafa, PhD

Academic Editor

PLOS ONE
---

## [Editor Report · Acceptance letter]

1 Feb 2023

PONE-D-22-20891R1 

Antimycotic effect of 3-phenyllactic acid produced by probiotic bacterial isolates against Covid-19 associated mucormycosis causing fungi  

Dear Dr. Bhukya:

I'm pleased to inform you that your manuscript has been deemed suitable for publication in PLOS ONE. Congratulations! Your manuscript is now with our production department. 

Kind regards, 

on behalf of

Dr. Ghulam Mustafa 

Academic Editor

PLOS ONE